# Influence of *Penicillium lanosum* and *Staphylococcus equorum* on Microbial Diversity and Flavor of Mianning Hams

**DOI:** 10.3390/foods13162494

**Published:** 2024-08-08

**Authors:** Wenli Wang, Yanli Zhu, Wei Wang, Jiamin Zhang, Daolin He, Lili Ji, Lin Chen

**Affiliations:** 1Key Laboratory for Meat Processing of Sichuan Province, Chengdu University, Chengdu 610100, China; 19983419913@163.com (W.W.); liza.yl.zhu@outlook.com (Y.Z.); wangwei8619@163.com (W.W.); jasminejjjjjj@163.com (J.Z.); jilili@cdu.edu.cn (L.J.); 2Chongqing General Station of Animal Husbandry Technology Extension, Chongqing 405400, China; hedaoling321@163.com

**Keywords:** Mianning ham, fermentation agent, microorganism, flavor, high-throughput sequencing technology

## Abstract

Mianning ham is a traditional meat product in China. In this experiment, solid-phase microextraction–gas chromatography (SPME-GC-MS) and high-throughput sequencing were used to study the effects of adding *Penicillium lanosum* and adding the mixture of *Penicillium lanosum* and *Staphylococcus equorum* on the flavor and microbiology of Mianning ham. The results showed that the addition of the ferments resulted in an increase in the abundance of both the dominant bacterial phylum (Thick-walled Bacteria) and the dominant fungal phylum (Ascomycota). The variety of volatile flavor substances and key flavor substances increased after adding fermentation agents. A free amino acid analysis showed that hams from the *Penicillium lanosum* and *Staphylococcus equorum* group had significantly higher umami flavor amino acids than the control group and *Penicillium lanosum* group. Therefore, inoculation with *Penicillium lanosum* and *Staphylococcus equorum* favored the dominant bacteria and flavor of Mianning ham.

## 1. Introduction

Mianning ham is a kind of dry-cured meat product whose production technology includes material selection, repair, curing, washing, drying, fermentation, and storage. It is a special meat product from Mianning County, Sichuan Province, that has a unique flavor and strong aroma. Fermentation agents are separate or mixed microbial cultures of known concentrations used to promote and carry out the fermentation of meat products. Bacteria, particularly lactic acid bacteria (LAB) and coagulase-negative staphylococci (CNS), as well as yeasts and molds [1,2], can be used as fermenters to help improve the safety of fermented meat products. In addition, fermenters may help standardize product characteristics and reduce the maturation time of fermented meat products [3].

Hu [4] et al. used *Staphylococcus xylosus* P2, a negative *staphylococcus*, as a fermenter. The color, aroma, texture, taste, and overall acceptability of fermented beef jerky were found to be improved. Zhu [5] et al. discovered that by adding fermenting agents, different flavor substances in ham increased during different stages of fermentation. Wang [6] et al. also experimentally found that *Staphylococcus* spp. are beneficial bacteria in fermented meat products and help to improve color and flavor. Zhou [7] et al. found that several fermenting agents from *Staphylococcus*, *Penicillium*, *Mucor*, *Micrococcus*, and *Lactobacillus* genera have shown great potential in preventing microbial hazards and improving the flavor quality of dry-cured ham through studying the mechanism of off-flavor formation and the potential control of fermentation.

During the processing of dry-cured ham, the microbial flora in the ham varies at different fermentation stages. He [8] et al. studied the microbial flora during the fermentation process of Jinhua ham and found that the microorganisms initially increase during fermentation and decrease when the ham is mature and fragrant, with predominant bacteria being *Staphylococcus* and lactic acid bacteria. *Penicillium* is dominant in the early stage of fermentation, while Aspergillus is dominant in the later stage. Important microbial species in ham include *lactic acid bacteria*, *Staphylococcus*, *Micrococcus*, *molds*, and *yeast* [9]. The instability of temperature and humidity during fermentation can also affect the quality and flavor of the meat by influencing the microorganisms [10]. This is due to the microbial degradation of proteins, fats, and carbohydrates [11,12] and the formation of acids, aldehydes, ketones, esters, hypoxanthines, and short peptides, etc., giving the ham its unique flavor and color [11,12,13]. However, the fermentation of these microorganisms is uncontrollable and highly unstable [14]. Therefore, microorganisms are one of the most important regulatory measures for ham quality formation [15], and changes in microorganisms can also lead to changes in flavor substances.

In this study, the effects of the addition of *Penicillium woollyum* and *Staphylococcus equorum* on bacterial community changes and volatile flavor substances during the fermentation process of Mianning ham were investigated.

## 2. Materials and Methods

### 2.1. Samples

The experimental samples were thirty-day-old naturally fermented hams from Jiuyuan Ham Factory, Mianning County, Sichuan Province. The samples were divided into a control group (RG), a monomicrobial fermentation group with vaccinations of *Penicillium lanosum* (YM), and a composite fermentation group with vaccinations of *Penicillium lanosum* and *Staphylococcus equorum* (YMMW). The samples were fermented separately for 270 days. The temperature was controlled at 22 °C and relative humidity at 60–70%. This study used *Penicillium lanosum* and *Staphylococcus equorum* isolated through traditional cultivation by Chen et al. [16].

According to the standard GB/T 4789.17-2003 “Food Hygiene Microbiology Testing Meat and Meat Products Testing” [17], samples were collected from the geometric center of Mianning ham fermented for 270 d using a sterile scalpel. The vacuum-packed samples were stored in a refrigerator at −80 °C for subsequent experiments.

### 2.2. High-Throughput Sequencing

All microbial DNA sequencing, PCR amplification, purification, and sequencing of ham samples were commissioned by OE Biotech Co., Ltd. (Shanghai, China).

Referring to Nossa C W [18] et al., the primers 343F:5′-TACGGRAGGCAGCAG-3′ and 798R:5′-AGGGTATCTAATCCT-3′ were used to amplify the V3-V3 rDNA of the bacterial 16S rDNA by combining the adaptor and barcode sequences. For the rDNA V3-V4 region, primers ITS1F:5′-CTTGGTCATTTAGAGGAAGTAA-3′ and ITS2:5′-GCTGCGTTCTTCATCGATGC-3′ combined with adapter sequences and barcode sequences were used to amplify fungal ITS regions [19].

The Amplicon Sequence Variant (ASV) classification of valid tags of sequences was conducted according to 97% similarity [20] using the microbiome analysis tool VSearch (Version 2.4.2) [21,22]. The raw image data files obtained from high-throughput sequencing were analyzed by Base Calling to be converted into raw sequencing sequences, which are called raw data. The results were stored in FASTQ file format, which contained the sequence information (reads) as well as their sequencing quality information. After the data were downloaded from the machine, the primer sequences of raw data were first cut using cutadapt 1.18 software; then, using DADA2, the double-ended raw data qualified in the previous step were quality-filtered according to the default parameters of QIIME 2 (2020.11), and reads containing unknown nucleotide (N) ≥ 10%, phred quality score ≤ 20, and bases ≥ 50% were deleted. Then, clean reads obtained after filtering were subjected to quality control analyses such as noise reduction, splicing, and chimera removal. The clean reads were merged into tags using FLSAH (version 1.2.11) according to the criteria of a maximum mismatch rate of 2% and a minimum overlap of 10 bp [23]. The low-quality tags were filtered to obtain clean tags [24] at a 97% similarity level. The clean tags were clustered into ASVs using UPARSE [25]. The tags were then checked for chimeras (for 16S sequencing). The filtered Effective Tags were then clustered, with the highest abundance as representative sequences of individual ASVs. Based on the obtained ASV abundance table, representative sequences of each ASV were selected for comparative annotation in the database using the QIIME 2 (2020.11) [26]. The 16S rDNA and ITS were aligned in Silva (version 138) and Unite databases, respectively. The representative sequence collections of fungi and bacteria were annotated for classification using the default parameters of the q2-feature-classifier 2019.1 software [27]. Microsoft Excel 2019 software was used for data statistics. IBM SPSS Statistics 22.0 software was used for data file processing, and Origin 2017 software, SIMCA 13.0 software, and R Studio R 3.5.1 software were used for image plotting. 

### 2.3. Determination of Volatile Flavor Substances

The volatile flavor substances was determined according to Bi et al. [28]. The three groups of ham samples were chopped and weighed accurately to 3.00 g into 15 mL headspace vials. Then, 1 μL of 2,4,6-trimethylpyridine was added as an internal standard to the headspace vials, and the headspace vials were sealed. The volatiles were extracted by headspace solid phase microextraction (SPME). The pretreatment conditions of the samples were set by the CTC autosampler: the temperature of the heating chamber (75 °C), the heating time (45 min), the sample extraction time (20 min), and the analysis time (5 min) were set.

GC conditions: HP-5ms-UI column (30 m × 0.25 mm, 0.25 mm) was used; the pressure was 32.0 kpa; the column flow rate was 1.0 mL/min; the carrier gas was helium for splitless injection. The temperature of the injection port was 250 °C.

Column temperature conditions: Started at 35 °C for 20 min, then increased to 200 °C at a rate of 5 °C/minute, and finally to 250 °C at a rate of 15 °C/minute. Held for 5 min.

Mass spectrometry conditions: An electron ionization (EI) source was used; the electron energy was 70 ev; the temperature of the ion source was 250 °C; the temperature of the transmission line was 150 °C; the mass scanning range was 35∼500 *m*/*z* with a scan rate of 1 scan/s. The detector voltage was 350 V.

Key volatile flavor substances: Calculated using the Odor Activity Value (OAV) with the formula shown in (1).
(1)$$OAV=\frac{A_{1}}{A_{2}}$$

*A*_1_ is the amount of the substance in the sample/(μg/kg); *A*_2_ is the odor threshold of the substance/(μg/kg).

### 2.4. Determination of Free Amino Acids

According to Lin et al. [29], the sample was hydrolyzed at 110 °C for 22 h, and then cooled and filtered. Set volume to50 mL and 1 mL of the sample was use and dried under reduced pressure (45 °C), dissolved in 1 mL of ultrapure water, dried again under reduced pressure, and then added to 1 mL of sodium citrate buffer solution. It was then filtered through a 0.22 µm aqueous membrane.

Derivatization: The above liquid to be tested was diluted to 200 µL with ultrapure water according to the concentration. Then, 200 µL of each derivatization reagent, S solution and Y solution, respectively, was added and left to stand at 40 °C for 1 h. Then, 800 µL of n-hexane was added, mixed, and left to stand and stratify. The lower layer of clear liquid was then removed after 10 min for testing.

Instrumental conditions: ODS column for amino acid analysis, column temperature 40 °C; wavelength 254 nm; flow rate 1 mL/min.

Key amino acids: Calculated using Taste Activity Value (TAV) with the formula shown in (2).
(2)$$TAV=\frac{B_{1}}{B_{2}}$$

*B*_1_ is the amount of the substance in the sample/(g/100 g); *B*_2_ is the threshold of the substance for odor/(g/100 g).

### 2.5. Data Analysis

Origin 2017 and Microsoft Excel 2019 (Microsoft, Redmond, WA, USA) were used for data statistics, while SIMCA 13.0 software and R language were used for image plotting. IBM SPSS Statistics 22.0 was used for the *t*-test of variance. Statistical analyses including one-way analysis of variance (ANOVA) and Fisher’s multiple range test were carried out using IBM SPSS Statistics 22.0 (IBM, Chicago, IL, USA). Principal component analysis (PCA) was performed using SIMCA-P+ 13.0 (Umetrics, Umea, Sweden) software.

## 3. Results and Analysis

### 3.1. High-Throughput Sequencing

#### 3.1.1. Microbial Diversity

From Table 1, we can see that the total number of bacterial ASVs was higher than that of fungal ASVs, indicating a higher species richness of bacteria in Mianning ham samples than fungi. In 16S rDNA, the RG group had the highest Shannon index (3.245), while the YM group had the lowest Shannon index (1.54), indicating that the RG group had the highest bacterial abundance. The YM group had the lowest bacterial abundance. The Chao1 (143.87) and ACE (143.50) indices of the YMMW group were the highest, indicating that the YM group had the most bacterial species. Shao [30] et al. and Xiao [31] found that fermenter cultures significantly reduced the diversity of bacterial communities in fermented meat products, which was consistent with these results.

#### 3.1.2. Differences in Microbial Communities

From Figure 1, it can be seen that the total number of bacteria detected in the ham samples of the RG group, YM group, and YMMW group was 569, of which 39 bacteria were common to the samples of the three groups. For the unique bacteria, the lower number in the ham samples with added fermentation agent may be due to the competitive growth of bacteria, where *Penicillium lanosum* and *Staphylococcus equorum* inhibit the growth and reproduction of other bacteria to a certain extent. A total of 51 fungi were detected in the three groups and 8 fungi were common to all three groups. The highest number of endemic fungi was found in the RG group (31), while the lowest number was found in the YMMW group (6). The reason for this may be that the moisture content decreased in the later stage of fermentation in the ham samples. At the same time, the salt content increased, which may have inhibited the growth and reproduction of some fungi in the Mianning ham [32]. 

#### 3.1.3. Species Distribution

The relative abundance of the top 15 bacterial phyla in the three groups is shown in Figure 2A. The relative abundance of Firmicutes (the dominant bacterial phylum) were 96.90% (YM group), 84.35% (YMMW group), and 70.35% (RG group). Firmicutes and Proteobacteria were the most dominant microbiota in all samples. The average relative abundance reached 85.3% and 10.4%, respectively, similar to reported findings for Norden ham. The dominant bacterial phyla in Norden ham during fermentation were Actinobacteria, Thick-walled Bacteria, Mycobacteria, and Ascomycetes, similar to the results of this study [33]. Chinese fermented sausage was also found to have the highest amount of thick-walled bacterial phyla among all treatments [30].

As shown in Figure 2B, the top 15 microbial bacterial genera in terms of abundance were identified at the genus level, such as *Staphylococcus*, *Cobetia*, and *Acinetobacter*. In the RG group, the dominant bacterial genera were *Staphylococcus* (65.47%) and *Cobetia* (16.17%). In the YM group, the dominant bacterial genus was *Staphylococcus* (96.26%). Among them, *Staphylococcus* was the dominant bacterial genus in all three groups, which produced lipase and protease in the fermented meat and could promote the formation of unique aroma and flavor during the fermentation and maturation of ham. At the late fermentation stage, the dominant genera in Xuanen ham were *Staphylococcus*, *Serratia*, and *Methylobacterium*, similar to Mianning ham [13]. The abundance of *Staphylococcus* was highest throughout the processing of Norden ham, similar to the results of this study. He [8] et al. studied the microflora of Jinhua ham during fermentation. They found that the main bacteria were *Staphylococci.*

As shown in Figure 3A, the top 15 microbial fungal phyla in terms of abundance were identified, such as Ascomycota, Basidiomycota, and Chytridiomycota. The dominant fungal phylum in the samples of three groups was Ascomycetes. The relative abundance of the YMMW group was significantly higher than that of other two groups. In Saba ham, the Ascomycetes phylum was the dominant fungal group among the surface and internal fungi of the ham, similar to the results of this study [34]. Wang [35] et al. used high-throughput sequencing to reveal that the dominant organisms at the phylum level were Ascomycetes and Thick-walled Bacteria in Mianning ham.

As can be seen from Figure 3B, in the RG group ham samples, the dominant fungal genera were *Aspergillus* (41.7%), *Yamadazyma* (35.12%), and *Debaryomyces* (18.65%). The dominant fungal genera in Panxian ham were *Aspergillus* and *Penicillium*, which were similar to the results of this study [36]. The *Yamadazyma* and *Aspergillus* are two of the major fungal communities on the surface and inside of Xuanwei hams [37,38]. In the ham samples of the YM group, the dominant fungi were *Debaryomyces* (74.38%) and *Aspergillus* (11.58%). In contrast, the dominant fungal genera in the samples from the YMMW group were *Debaryomyces*, *Aspergillus,* and *Yamadazyma,* which were the dominant fungal genera in all three groups of samples, with the highest relative abundance in the YM group and the lowest relative abundance in the RG group. Wan [39] isolated and screened *Debaryomyces* in Xuanwei ham.

As shown by the heatmap of bacterial community abundance differences (Figure 4A), Proteobacteria, Bacteroidota, Desulfobacteria, Actinobacteriota, Fusobacteria, Deferribacterota, Fusobacteriota, and Deferribacterota had higher relative abundance in the RG group. In the YM group, the relative abundance of Firmicutes, Campilobacterota, and Bdellovibrionota were higher in bacterial community. In the YMMW group, the relative abundance of Acidobacteria was higher. Studies have shown that Proteobacteria and Firmicutes were the dominant bacteria during the processing of Nuodeng ham and Panxian ham [40,41].

As can be seen in Figure 4B, bacteria at the genus level in the RG group with the highest relative abundance were Cobetia, Acinetobacter, Muribaculaceae, Ruminococcus, Psychrobacter, Clade_Ia, Bacteroides, Lentibacillus, [Eubacterium]_coprostanoligenes_group, Butyricimonas, and Lachnospiraceae_NK4A136_group. In the YM group, the only genus with the highest relative abundance was Staphylococcus, which plays an important role in the quality and color during ripening [4,42]. In the YMMW group, Pseudomonas, Lactobacillus, and Parabacteroides had higher relative abundance. Zhou [7] et al. found that several fermenting agents from the Staphylococcus, Penicillium, Mucor, Micrococcus, and Lactobacillus genera have shown great potential in preventing microbial hazards and improving the flavor quality of dry-cured ham through studying the mechanism of off-flavor formation and the potential control of fermentation.

As shown by the heatmap of fungal community abundance differences (Figure 5A), the relative abundance of Basidiomycota at the phylum level was the highest in the samples from the RG group. In the YM group, no fungal phylum with high relative abundance was detected except Others. In the YMMW group, at the phylum level, Ascomycota and Chytridiomycota were higher than in other two groups, which was consistent with the results of other studies [40,43].

From Figure 5B, it can be seen that at the genus level, the control samples of *Aspergillus*, *Yamadazyma*, *Wallemia*, *Rhodotorula*, *Mrakia*, *Pleurotus*, *Plicaturopsis*, *Mycosphaerella*, and *Leptospora* had the highest relative abundance in the RG group. Studies have shown that *Aspergillus* was the dominant fungal genus of Jinhua ham [44], which was similar to the results of this study. In the YM group, *Debaryomyces*, *Candida,* and *Microascus* had the highest relative abundance. In the YMMW group, *Trichoderma* had the highest relative abundance.

### 3.2. Volatile Flavor Substances

Table 2 shows that the total absolute content of flavor compounds was the highest in the YM group, reaching 11,575.58 μg/kg, followed by the RG group (10,857.42 μg/kg). The total absolute content of compounds was the lowest in the YMMW group (9557.37 μg/kg).

Twenty-two key flavor substances that contributed significantly to the overall flavor profile of Mianning ham were screened out though OAV value calculation (Table 3). Among the 22 key flavor substances, the most prominent contributors were mainly aldehydes (13), followed by alcohols (5). Aldehydes generally had lower thresholds and contributed more to the flavor of Mianning ham.

Hexanal and nonanal are the most abundant aldehydes in many dry-cured hams. Hexanal has a grassy taste, while heptanal octanal and nonanal have a fatty aroma [45]. The contents of heptanal octanal and nonanal in the YMMW group were higher than those in the YM group and RG group, indicating that ham fermented with *Penicillium lanosum* and *Staphylococcus equorum* has a stronger fat aroma after maturity. However, when the concentration of aldehydes is large, it may cause the ham to produce an unpleasant odor [46]. Benzaldehyde (bitter almond odor) and phenylacetaldehyde (acorn odor) are often detected in dry-cured hams, and in Iberian hams, these two aldehydes are used as active odorant ingredients [47]. In this experiment, benzaldehyde and phenylacetaldehyde were only detected in the YM group and YMMW group, indicating that Mianning ham matured using starter fermentation has richer odor components than ham matured by natural fermentation, and a microbial starter can provide better flavor.

2-heptanone has a fruity aroma similar to pear, and the ham from the YMMW and YM groups had a fruitier aroma and better flavor than the ham from the RG group. 2-nononone was only detected in YMMW ham (204.35 ± 23.65 μg/kg). 2-nononone has a soapy odor, which may be related to the formation of odor in the late storage of ham. 1-octene-3-ol has been detected in Rugao ham, Panxian ham, and Jinhua ham. Similar to the results of this experiment, 1-octene-3-ol endows dry-cured ham with fragrance and a vegetable aroma. It is often used as a characteristic flavor substance [48,49,50]. Hexanol imparts a loose flavor to meat products, indicating that ham from the YM group has more rosin odor after maturity. Heptanol imparts a fresh aroma to ham, which was detected in the YMMW group, YM group, and RG group ham. Hexanol and heptanol are the characteristic substances of fermented 90 d Dahe black pig ham, which are similar to the results of this study [51]. Linalool belongs to the chain terpene group and has a woody flavor with a low threshold, which was detected in the YMMW group and YM group, indicating that it contributed more to the flavor of ham in the fermentation group [52].

As can be seen from Figure 6, the PCA was able to distinguish the volatile flavor substances in the RG group, the YM group, and the YMMW group. Among them, the volatile flavor substances of the RG group and the YM group were more similar in the maturation stage compared with the YMMW group.

### 3.3. Free Amino Acid

As can be seen from Table 4, a total of 16 free amino acids were detected in the three groups of ham, including 2 umami amino acids, 8 sweet amino acids and 6 bitter amino acids. The umami amino acids in the three groups of ham were mainly glutamic acid and aspartic acid. The total amino acid content of ham in the YMMW group was the highest (28.34 g/100 g), followed by the YM group (27.15 g/100 g), and the total amino acid content of ham in the RG group was the lowest (25.39 g/100 g).

The aroma of ham is not only closely related to amino acids, but also depends on the type of amino acid and the threshold of flavor presentation [53]. As shown in Table 5, these 12 amino acids had a greater contribution to the flavor of the hams in the RG group and YM group, with histidine, arginine, tyrosine, isoleucine, leucine, and phenylalanine being closely related to the bitter flavor in Mianning ham. In the YMMW group, glutamic acid, histidine, lysine, and alanine had a greater contribution to the overall flavor presentation of the YMMW group. Among them, glutamic acid and histidine had lower threshold values, which had an important effect on the flavor formation of Mianning ham. Among the three groups of ham samples, glutamic acid had a higher TAV value than the other amino acids, which contributed the most to the umami flavor in Mianning ham. Glutamic acid, however, also has an acidic taste when reacting with sodium salt. It not only promotes the umami taste of food, but also buffers the salty and acidic tastes [54]. It has been shown that glycine, phenylalanine, aspartic acid, alanine, and glutamic acid contribute to the formation of the unique fermented flavor in hams [55]. Lysine is the main sweet amino acid in all three groups of hams and contributes significantly to the flavor of Mianning ham. Table 5 shows that the TAV of glutamate and lysine in the YMMW group was greater than that of the YM group than that of the RG group; thus, the flavor of the YMMW group was superior, followed by the YM group.

Twelve key flavor-presenting amino acids were screened according to the TAV method. The content heatmaps of amino acids with TAV values greater than or equal to 1 in the three groups of ham samples were plotted using the R language (Figure 7). The closer the color to red in the graph, the higher the level of the amino acid. The closer the color to blue, the lower the level of the amino acid.

As can be seen from Figure 7, in the RG group, the content of leucine and phenylalanine was higher, resulting in a bitter taste, and the bitter taste contributed more to the characteristic flavor of the Mianning ham. In the YM group, the contents of arginine and histidine were higher, and the bitter taste contributed more to the flavor. The YMMW group contained alanine, valine, glycine, methionine, and aspartic and glutamic acid, which were sweet. The contents of alanine, valine, glycine, glutamic acid, and methionine were the highest. This sweetness contributed more prominently to the YMMW group. The flavor of ham in the YMMW group was significantly better than that in the YM group and RG group. The lipase and protease produced by *Staphylococcus* in the fermentation of meat can promote the formation of the distinctive aroma and taste of the ham in the process of fermentation and ripening. This is due to the secretion of proteases by *Staphylococcus*, which hydrolyze proteins to form free amino acids and peptides [56]. The relative abundance of *Staphylococcus* was higher in the YM and YMMW groups than in the RG group, suggesting that inoculation with the fermenter had an ameliorating effect on proteolysis and flavor. Niu [57] reached a similar conclusion.

## 4. Conclusions

In this study, it was found that the flavor of ham became better with the addition of *Penicillium lanosum* (YM) and its mixed fermenter with *Staphylococcus equorum* (YMMW). Analyzed in terms of both volatile flavor substances and free amino acids, the YMMW group had better flavor than the YM group and the RG group. In addition, the relative abundance of dominant bacteria increased after the addition of the fermenter, and the growth of stray bacteria was inhibited. The relative abundance of *Staphylococcus* was also the highest in the YMMW group, followed by the YM group. In conclusion, the addition of *Penicillium lanosum* and its mixed fermenter with *Staphylococcus equorum* as fermentation agents is conducive to improving the flavor of ham.

## Figures and Tables

**Figure 1 foods-13-02494-f001:**
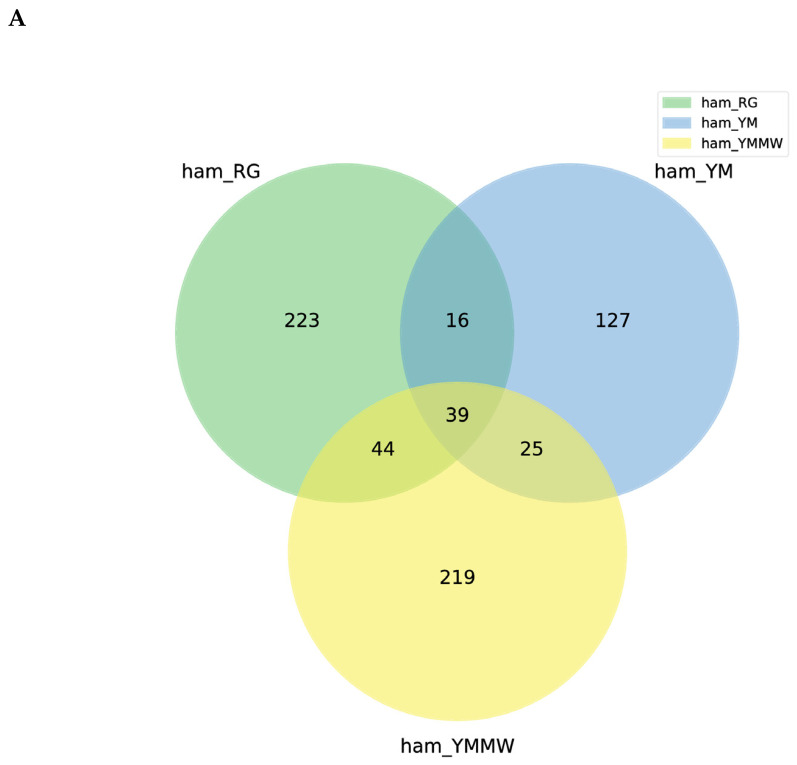
Venn diagram based on ASVs at the phylum (**A**) level and genus (**B**) level.

**Figure 2 foods-13-02494-f002:**
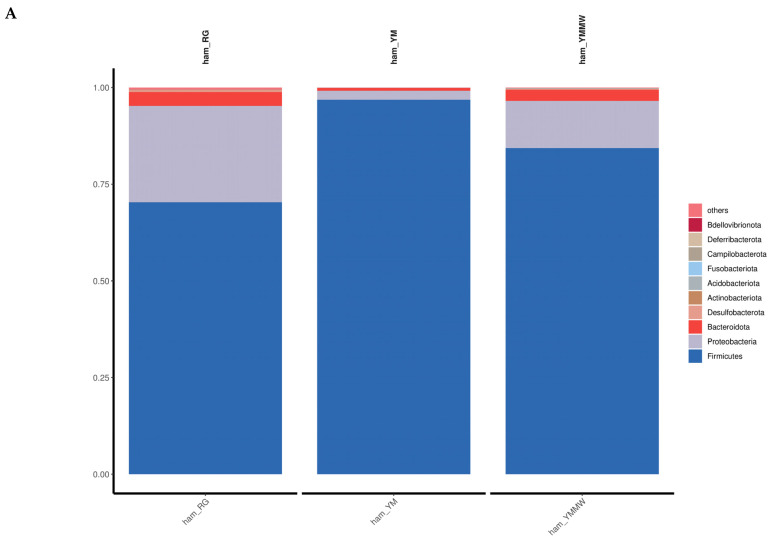
The relative abundance between bacteria groups at the phylum (**A**) level and genus (**B**) level.

**Figure 3 foods-13-02494-f003:**
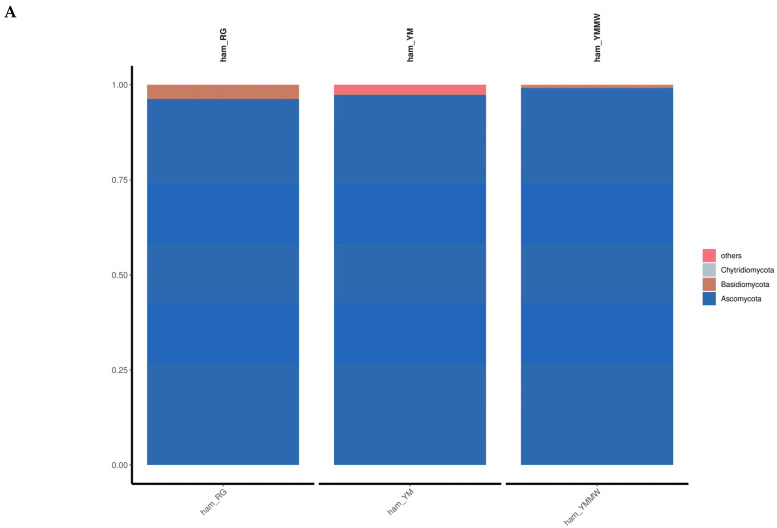
The relative abundance between fungi groups at the phylum (**A**) level and genus (**B**) level.

**Figure 4 foods-13-02494-f004:**
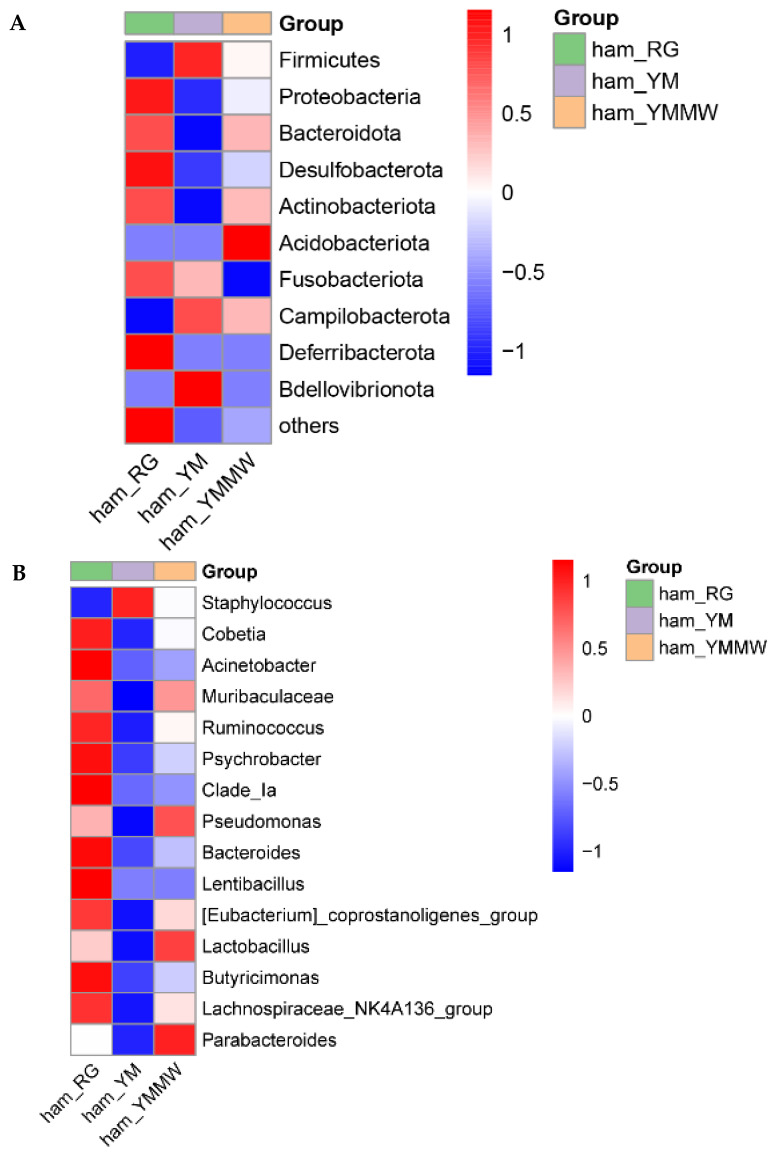
Heatmap of differences in bacterial community abundance at the phylum (**A**) level and genus (**B**) level.

**Figure 5 foods-13-02494-f005:**
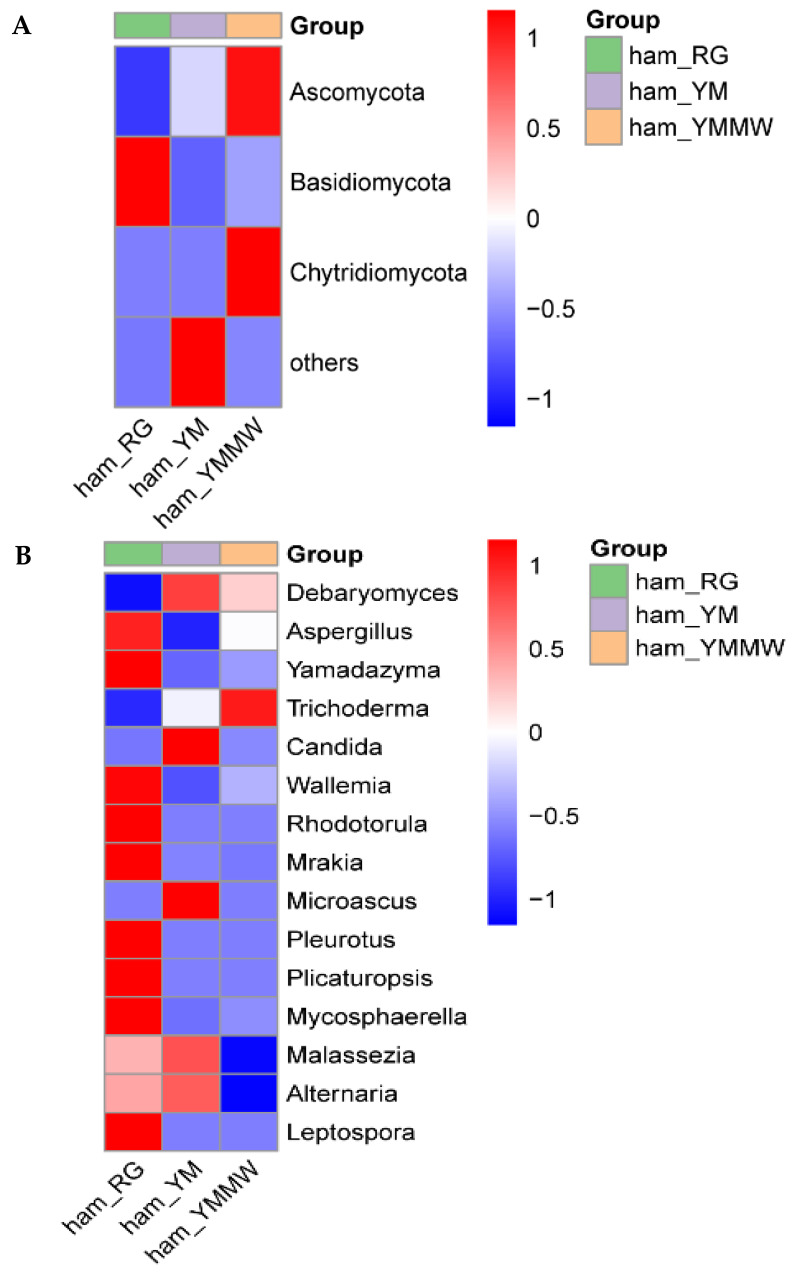
Heatmap of differences in abundance of fungal communities at the phylum (**A**) level and genus (**B**) level.

**Figure 6 foods-13-02494-f006:**
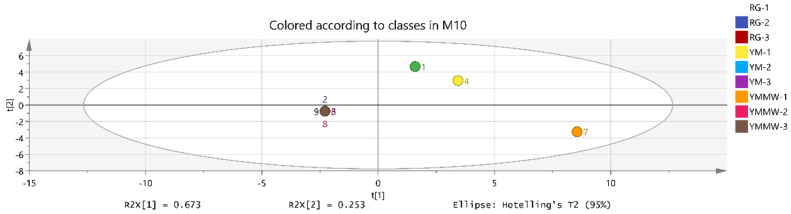
Plot of principal component analysis scores of flavor substances of Mianning ham at fermentation maturation stage. The numbers 1-3 stand for RG-1, RG-2, RG-3, 4-6 for YM-1, YM-2, YM-3,7-9 for YMMW-1, YMMW-2, YMMW-3.

**Figure 7 foods-13-02494-f007:**
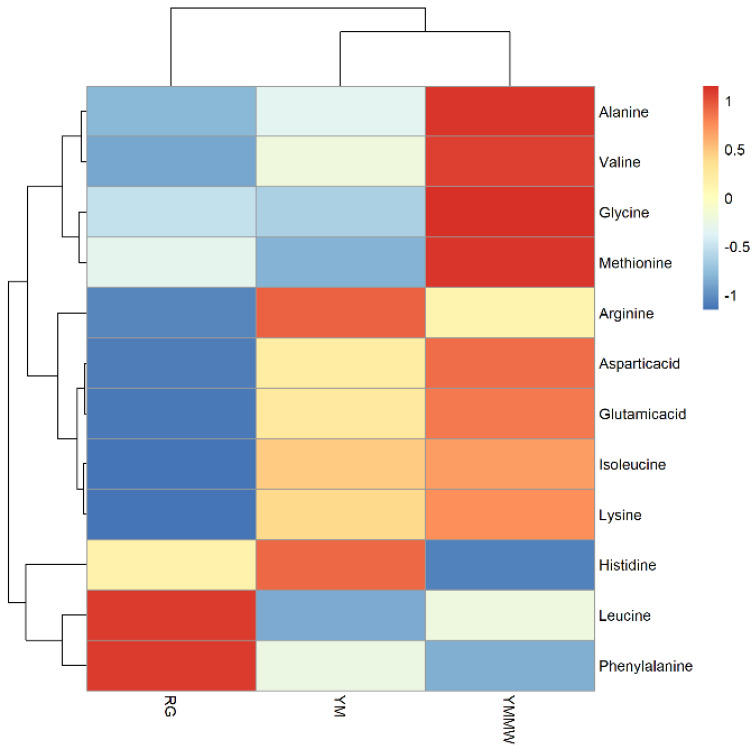
Heatmap of differences in key amino acid content.

**Table 1 foods-13-02494-t001:** Microbial diversity index of Mianning ham.

Samples		Observed OTUs	Shannon	Chao1	Simpson	ACE	Sequence Coverage
RG	Bacterial 16S rDNA	135.63	3.245	135.63	0.66	135.36	0.99
	Fungal ITS	20.67	2.26	20.67	0.72	26.5	1
YM	Bacterial 16S rDNA	92.63	1.54	92.74	0.36	92.67	0.99
	Fungal ITS	14.3	1.277	14.3	0.4	14	0.99
YMMW	Bacterial 16S rDNA	143.3	2.93	143.87	0.61	143.5	0.99
	Fungal ITS	11.33	1.68	11.33	0.6	13.5	0.99

**Table 2 foods-13-02494-t002:** Statistics of volatile flavor substances in Mianning ham.

Chemical Compound	Absolute Content/(μg/kg)			
	RG	YM	YMMW	Total
Aldehyde	5291.94	6661.41	5013.34	16,966.69
Ketone	149.62	208.88	543.36	901.86
Acids	486.02	431.8	225.61	1143.43
Salts	1023.83	430.23	627.97	2082.03
Alcohol	2169.96	1700.73	1122.37	4993.06
Hydrocarbons	796.82	636.09	868.29	2301.2
Others	939.23	1506.44	1156.33	3602
Total	10,857.42	11,575.58	9557.37	31,990.37

**Table 3 foods-13-02494-t003:** OAV of volatile flavor substances in the fermentation maturation stage of Mianning ham.

Chemical Compound	Threshold (μg/kg)	OAV Value (OVA ≥ 1)		
		RG	YM	YMMW
hexanal	7.50	357.83	199.55	115.49
heptanal	10.00	23.769	18.80	24.78
3-Methylthiopropionaldehyde	0.04	-	-	532.75
benzaldehyde C6H5CHO, the simplest aromatic aldehyde	50.00	-	1.57	4.33
octanal	0.10	3318.90	3855.30	5535.60
phenylacetaldehyde (PAG)	9.00	-	3.97	14.95
nonanal	3.50	327.17	342.08	468.43
trans-2-Nonanal	0.07	2507.00	1616.86	1430.43
decanal	0.90	59.68	94.28	318.33
trans,trans-2,4-Nonadienal	0.06	663.50	423.00	212.67
undecanal	14.00	-	0.82	1.82
trans,trans-2,4-Decadienal	0.03	1359.00	1275.00	1005.67
dodecanal	1.07	-	13.34	30.71
2-Heptanone	70.00	-	1.55	2.84
2-Nonanone	25.00	-	-	7.09
2-Undecanone	10.00	-	-	2.62
hexanoic acid	200.00	1.36	0.37	0.05
hexanol	200.00	0.16	1.07	0.69
1-Octen-3-ol	2.00	213.32	297.59	137.27
octanol	54.00	-	4.28	-
linalool	1.50	-	11.03	65.96
nonanal	2.00	-	-	15.27

**Table 4 foods-13-02494-t004:** Free amino acid content of Mianning ham at fermentation maturation stage.

Amino Acid Name	Flavor Properties	Amino Acid Content/(g/100 g)		
		RG	YM	YMMW
Aspartic (Asp)	Umami	2.42	2.69	2.83
Glutamic (Glu)	Umami	3.57	4.13	4.36
Threonine (Thr)	Sweetness	1.39	1.31	1.26
Serine (Ser)	Sweetness	1.14	1.17	1.04
Glycine (Gly)	Sweetness	1.22	1.21	1.35
Histidine (His)	Bitterness	1.38	1.54	1.13
Arginine (Arg)	Bitterness	1.55	1.87	1.74
Alanine (Ala)	Sweetness	1.42	1.78	2.97
Proline (Pro)	Sweetness	1.16	1.21	1.01
Tyrosine (Tyr)	Bitterness	0.98	0.96	0.87
Valine (Val)	Sweetness	1.28	1.33	1.42
Methionine (Met)	Sweetness	0.73	0.64	0.98
Isoleucine (Ile)	Bitterness	1.20	1.35	1.37
Leucine (Leu)	Bitterness	2.44	2.32	2.36
Phenylalanine (Phe)	Bitterness	1.25	1.16	1.12
Lysine (Lys)	Sweetness	2.26	2.48	2.53
Total amino acids (TAAs)		25.39	27.15	28.34

**Table 5 foods-13-02494-t005:** TAV values of amino acids in the fermentation maturation stage of Mianning ham.

Amino Acid Name	Threshold (mg/g)	TAV Value (TAV ≥ 1)		
		RG	YM	YMMW
Aspartic	1	2.42	2.69	2.83
Glutamic	0.3	11.9	13.77	14.53
Glycine	1.3	0.94	0.93	1.04
Histidine	0.2	6.9	7.7	5.65
Arginine	0.5	3.1	3.74	3.48
Alanine	0.6	2.37	2.97	4.95
Valine	0.4	3.2	3.33	3.55
Methionine	0.3	2.43	2.13	3.27
Isoleucine	0.9	1.33	1.5	1.52
Leucine	1.9	1.28	1.22	1.24
Phenylalanine	0.9	1.39	1.29	1.24
Lysine	0.5	4.52	4.96	5.06

## Data Availability

The original contributions presented in the study are included in the article, further inquiries can be directed to the corresponding author.

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
