# Peer review of "Influence of *Penicillium lanosum* and *Staphylococcus equorum* on Microbial Diversity and Flavor of Mianning Hams"

_foods, 2024, doi:10.3390/foods13162494_

Round 1

Reviewer 1 Report

Comments and Suggestions for Authors

The article titled "Influence of Penicillium Lanosum and Staphylococcus Equorum on microbial diversity and flavor of Mianning hams" investigated the effect of adding fermentation agents on the changes in bacterial community and volatile flavor compounds during the ham fermentation process. Despite of being intriguing in title, there are many issues should be address before further consideration. Abstract doesn't reflect your work well. The introduction is too short. The authors only reporting their data but not discussing them. They should more focus on discussing their findings. 

My specific comment are given in the pdf file.

Thank you 

Comments on the Quality of English Language

Author Response

Thank you very much for taking your valuable time to review this manuscript. All your comments and suggestions are greatly appreciated. We have carefully considered them and have endeavoured to address each and every one of them. Below is a list of changes we have made to the article based on your suggestions. Detailed change tracking is visible in Word document review mode.

Reviewer 1

Thank you very much for your professional comments on my article. As a result of your concerns, I need to answer a few questions and the answers are listed below. In the meantime, I have uploaded the revised manuscript

Q1: The treatment and methodology are not yet known.

I have placed this section in the ‘Sample Treatment’ section.

Q2: What does RG stand for?

RG stands for Mianning ham treated by conventional fermentation without inoculation of fermentation agent, which is used as a control group.

Q3: What about a group using Coccidioides alone?

Thank you for finding this out! We were just trying to understand the effect of the compounding action when adding another group of bacteria. So only one group was used.

Q4: Where is the discussion? You should discuss and detail your results. It is not enough to just report your results. (Table 3)

There is no discussion of Table 3 as it is only used as data support for Figure 6 and I have added a discussion of Table 3, thank you.

Q5: What about the sensory analysis test? Don't you think it would improve your article?

Thank you for your suggestion, this article focuses on the effect of the fermenter on the subjectivity of Mianning hams, therefore no sensory aspects have been added and your suggestion will be taken into serious consideration.

I will upload the final version.

Thank you for your comments and suggestions.

Reviewer 2 Report

Comments and Suggestions for Authors

English:
In some places, very long sentences are difficult to read.

2.1. Samples

Standard GB/T 4789.17-2003 has been replaced please check the compliance with the current standard.

Please provide a detailed description of the sample handling procedure.

Fermentation conditions?

3.1.2

Please consider inserting a clearer diagram. Ensure all labels are legible, with an appropriate font size and style. Avoid overly complex or decorative fonts.

150  - Please consider – the use term: "Total number of bacteria" is a general term referring to the quantity of all bacteria present in a sample or study. Please check the new literature positions like : Regueira-Iglesias A, Balsa-Castro C, Blanco-Pintos T, Tomás I. Critical review of 16S rRNA gene sequencing workflow in microbiome studies: From primer selection to advanced data analysis. Mol Oral Microbiol. 2023 Oct;38(5):347-399. doi: 10.1111/omi.12434. Epub 2023 Oct 7. PMID: 37804481.

 Sentence 159, why is it highlighted in yellow?

3.1.3. Species distribution

Please consider inserting a clearer diagram and more readable – This comment applies to all diagrams.

Describe how these results were interpreted to reach the conclusions stated.

Please keep the “Research Manuscript Sections” accordingly to Journal requirements! (https://www.mdpi.com/journal/foods/instructions#manuscript)

Discussion: Authors should discuss the results and how they can be interpreted in perspective of previous studies and of the working hypotheses.

The findings and their implications should be discussed in the broadest context possible, and the limitations of the work should be highlighted. Future research directions may also be mentioned.

This section may be combined with Results; however, I believe that, given the extensive nature of the study, the discussion should be presented in a separate section and subjected to a detailed analysis, taking into account the latest literature in this field.

Conclusions: This section is not mandatory but can be added to the manuscript if the discussion is unusually long or complex.

Funding: All sources of funding of the study should be disclosed. Clearly indicate grants that you have received in support of your research work

Comments on the Quality of English Language

English:
In some places, very long sentences are complex to read.

2.1. Samples

Standard GB/T 4789.17-2003 has been replaced. Please check the compliance with the current standard.

Please provide a detailed description of the sample handling procedure.

Fermentation conditions?

3.1.2

Please consider inserting a clearer diagram. Ensure all labels are legible, with an appropriate font size and style. Avoid overly complex or decorative fonts.

150  - Please consider the use of the term "Total number of bacteria," which is a general term referring to the quantity of bacteria present in a sample or study.

Please check the new literature positions like fo example Regueira-Iglesias A, Balsa-Castro C, Blanco-Pintos T, Tomás I. Critical review of 16S rRNA gene sequencing workflow in microbiome studies: From primer selection to advanced data analysis. Mol Oral Microbiol. 2023 Oct;38(5):347-399. doi: 10.1111/omi.12434. Epub 2023 Oct 7. PMID: 37804481.

Sentence 159, why is it highlighted in yellow?

3.1.3. Species distribution

Please consider inserting a clearer diagram that is more readable – This comment applies to all diagrams.

Describe how these results were interpreted to reach the conclusions stated.

Please keep the “Research Manuscript Sections” accordingly to Journal requirements! (https://www.mdpi.com/journal/foods/instructions#manuscript)

Discussion: Authors should discuss the results and how they can be interpreted from the perspective of previous studies and of the working hypotheses.

The findings and their implications should be discussed in the broadest context possible, and the limitations of the work should be highlighted. Future research directions may also be mentioned.

This section may be combined with Results however I believe that, given the extensive nature of the study, the discussion should be presented in a separate section and subjected to a detailed analysis, taking into account the latest literature in this field..

Conclusions: This section is not mandatory but can be added to the manuscript if the discussion is unusually long or complex.

Funding: All sources of funding for the study should be disclosed. Indicate grants that you have received in support of your research work

Author Response

Reviewer 2

I would like to thank you very much for your professional comments on my article. As your concern, there are several questions that I need to answer, and they are answered as follows.

Q1:. In some places, very long sentences are difficult to read.

Thank you for the suggestion, I've revised the long sentence

Q2:Standard GB/T 4789.17-2003 has been replaced please check the compliance with the current standard.

The microbiological test standards for meat and meat products are shown in the figure.Standard GB/T 4789.17-2003 has not been completely abolished and the new standard will be implemented next month.

Q3:Please provide a detailed description of the sample handling procedure.Fermentation conditions?

Thanks for the suggestion, I've added this section to the sample processing section

Q4: Please consider inserting a clearer diagram. Ensure all labels are legible, with an appropriate font size and style. Avoid overly complex or decorative fonts.

I've received your suggestions and made the changes.

Q5: 150  - Please consider – the use term: "Total number of bacteria" is a general term referring to the quantity of all bacteria present in a sample or study.

When using the term Total number of bacteria, I chose the case where all samples were included, e.g. the total number of bacteria in four samples.

Q6: Please check the new literature positions like : Regueira-Iglesias A, Balsa-Castro C, Blanco-Pintos T, Tomás I. Critical review of 16S rRNA gene sequencing workflow in microbiome studies: From primer selection to advanced data analysis. Mol Oral Microbiol. 2023 Oct;38(5):347-399. doi: 10.1111/omi.12434. Epub 2023 Oct 7. PMID: 37804481.

Thanks for the suggestion, I have read the literature and made some change.

Q7:  Sentence 159, why is it highlighted in yellow?

Thank you for your discovery. It's a mistake.

Q8: Please consider inserting a clearer diagram and more readable – This comment applies to all diagrams.

I've received your suggestions and made the changes.

Q9: Describe how these results were interpreted to reach the conclusions stated.

Thank you for your suggestions, I have made changes to the article.

Q10: Please keep the “Research Manuscript Sections” accordingly to Journal requirements! (https://www.mdpi.com/journal/foods/instructions#manuscript)

Discussion: Authors should discuss the results and how they can be interpreted in perspective of previous studies and of the working hypotheses.

The findings and their implications should be discussed in the broadest context possible, and the limitations of the work should be highlighted. Future research directions may also be mentioned.

This section may be combined with Results; however, I believe that, given the extensive nature of the study, the discussion should be presented in a separate section and subjected to a detailed analysis, taking into account the latest literature in this field.

Conclusions: This section is not mandatory but can be added to the manuscript if the discussion is unusually long or complex.

Thank you for your suggestions, I've made a lot of changes to the content of the article as you suggested

Q11:Funding: all sources of funding for the research should be disclosed. Clearly state the grants you have received to support your research work

This paper is not funded or sponsored

Thank you for your comments and suggestions.

Kind regards

Wenli Wang

Round 2

Reviewer 1 Report

Comments and Suggestions for Authors

To whom it may concern

The authors have clearly replied my comments and

the manuscript can now be further considered.

Author Response

Q1:作者已经明确回复了我的意见,现在可以进一步考虑手稿了。

感谢您审阅我的手稿并提出您的专业建议。

Reviewer 2 Report

Comments and Suggestions for Authors

Ok Good job

Author Response

感谢您审阅我的手稿并提出您的专业建议。